# Atomic-level polarization in electric fields of defects for electrocatalysis

Jie Xu [1,8], Xiong-Xiong Xue [2,8], Gonglei Shao [3] ✉, Changfei Jing[4], Sheng Dai [4], Kun He[1], Peipei Jia[5], Shun Wang [1], Yifei Yuan [1] ✉, Jun Luo [5] ✉ & Jun Lu [6,7] ✉

The thriving field of atomic defect engineering towards advanced electrocatalysis relies on the critical role of electric field polarization at the atomic scale. While this is proposed theoretically, the spatial configuration, orientation, and correlation with specific catalytic properties of materials are yet to be understood. Here, by targeting monolayer $MoS_2$ rich in atomic defects, we pioneer the direct visualization of electric field polarization of such atomic defects by combining advanced electron microscopy with differential phase contrast technology. It is revealed that the asymmetric charge distribution caused by the polarization facilitates the adsorption of H*, which originally activates the atomic defect sites for catalytic hydrogen evolution reaction (HER). Then, it has been experimentally proven that atomic-level polarization in electric fields can enhance catalytic HER activity. This work bridges the long-existing gap between the atomic defects and advanced electrocatalysis by directly revealing the angstrom-scale electric field polarization and correlating it with the as-tuned catalytic properties of materials; the methodology proposed here could also inspire future studies focusing on catalytic mechanism understanding and structure-property-performance relationship.

The utilization of renewable energy sources to solve the global fossil fuel crisis necessitates the process of energy conversion and storage[1]. This process involves interphase reactions and typically requires the presence of electrocatalysis and thus advanced electrocatalysts[2–5]. With the design of such materials delving into nano-, subnano- and atomic-scale strategies nowadays, understanding the atomic-level origins of electrocatalytic properties is crucial for developing efficient and low-cost catalysts. In fact, the essence of the electrocatalytic processes is the charge transfer between reactants and catalysts, which enables the adsorption of reactants and their successive activation/transformation[6]. The charge transfer efficiency of electrocatalysts is determined by the intrinsic field/charge distribution surrounding these surface atomic sites. However, such experimental studies have been scarcely reported due to the limit of appropriate characterization that is sensitive to atomic electric field.

A large number of advanced electrocatalysts have atomic defect structures on their surfaces. These defects can alter the electric field/charge distribution of electrocatalysts with enhanced catalytic performances[7–13]. More importantly, introducing atomic defects on the electrocatalyst surface leads to the generation of non-periodic electric fields (eg., polarized electric fields) that can significantly enhance the electric field effect at reaction interfaces, offering a

[1]College of Chemistry and Materials Engineering, Wenzhou University, Wenzhou, Zhejiang 325035, China. [2]School of Physics and Optoelectronics, Xiangtan University, Xiangtan 411105, China. [3]Interdisciplinary Research Center for Sustainable Energy Science and Engineering (IRC4SE2), School of Chemical Engineering, Zhengzhou University, Zhengzhou 450001, China. [4]Feringa Nobel Prize Scientist Joint Research Centre, School of Chemistry and Molecular Engineering, East China University of Science & Technology, Shanghai 200237, China. [5]ShenSi Lab, Shenzhen Institute for Advanced Study, University of Electronic Science and Technology of China, Longhua District, Shenzhen 518110, China. [6]College of Chemical and Biological Engineering, Zhejiang University, Hangzhou 310027, China. [7]Quzhou Institute of Power Battery and Grid Energy Storage, Quzhou, Zhejiang 324000, China. [8]These authors contributed equally: Jie Xu, Xiong-Xiong Xue. ✉e-mail: shaogonglei@zzu.edu.cn; yifeiyuan@wzu.edu.cn; jluo@uestc.edu.cn; junzoelu@zju.edu.cn

viable strategy to tune catalytic reaction kinetics[14–16]. Analyzing the non-periodic electric field on the electrocatalyst with high spatial accuracy is crucial for understanding the catalytic mechanism. Unfortunately, since the technical challenges of atomic imaging currently hinder the characterization of the non-periodic electric fields surrounding specific atomic defects, understanding such microscopic mechanisms largely relies on theoretical calculations[14–16]. This difficulty is leading to the inability to study the structure-performance relationship between the atomic electric fields of most catalysts with defects and catalytic activity.

Electrocatalytic HER is currently a research focus due to its high-energy density and clean advantages of hydrogen energy that offer an advanced solution for energy sustainability[2–5]. Among the many investigated catalysts, molybdenum disulfide ($MoS_2$) stands out not only because of its low cost and high efficiency but also due to its high two-dimensional (2D) structural variability, tuned by atomic defect engineering, to achieve effective property regulation[17–20]. Therefore, $MoS_2$, with point defects as a catalyst, is an ideal material system to explore the effect of electric field polarization of atomic defect sites on the as-tuned catalytic property and performance.

Herein, we realize controlled preparation of antisite defect structures (two S or single S atoms occupy the position of Mo atoms, denoted as $S2_{Mo}$-$MoS_2$ or $S_{Mo}$-$MoS_2$) with a concentration of 4 % in monolayer $MoS_2$ base planes by $H_2$/Ar atmosphere-assisted calcination. More importantly, we visualized the distribution of the polarized electric field surrounding the antisite defects by the atomic-resolution differential phase contrast (DPC) technology. Furthermore, combined with micro-reactor electrochemical testing, the excellent HER catalytic activity of the antisite defects is demonstrated. These results indicate that a polarized electric field significantly enhances catalytic activity.

The structure-performance relationship between the electric field polarization of the antisite defects and their as-tuned catalytic performance is well understood.

## Results

### Synthesis and characterization

In this work, we first obtained pristine monolayer 2D $MoS_2$ by chemical vapor deposition (CVD)[18]. As shown in Fig. 1a, the as-prepared $MoS_2$ sample was calcined at 400 °C in $H_2$/Ar atmosphere (5% $H_2$ concentration) for 1 and 5 min, to obtain 2D $MoS_2$ containing Mo vacancies (denoted as $V_{Mo}$-$MoS_2$-1) and antisite defects (denoted as $S2_{Mo}$-$MoS_2$-5), respectively. Energy-dispersive X-ray spectroscopy (EDS) elemental maps (Supplementary Fig. 1) proved the existence of S and Mo elements, indicating that pristine $MoS_2$ was synthesized. Atomic force microscopy (AFM) measurements show that the thickness of the pristine $MoS_2$ is 0.84 nm (Supplementary Fig. 2), which conforms to the characteristics of monolayer $MoS_2$[19]. Further, the optical images in Fig. 1b-d also show that the three $MoS_2$-based materials all have regular triangle shapes. In addition, Raman and photoluminescence (PL) spectroscopy were used to investigate the spectral feedback information caused by the structural changes in the three $MoS_2$-based materials. The Raman spectra (Supplementary Fig. 3a) show that compared with the pristine $MoS_2$, the broadening of the peak spread of $V_{Mo}$-$MoS_2$-1 and $S2_{Mo}$-$MoS_2$-5 indicates that they have more defect structures[18,20]. In the PL spectra (Supplementary Fig. 3b), for $V_{Mo}$-$MoS_2$-1 and $S2_{Mo}$-$MoS_2$-5, the intensities of the main peaks are suppressed and the positions of the main peaks show a red shift, compared to the pristine $MoS_2$. These findings further confirm that there are point defects in $V_{Mo}$-$MoS_2$-1 and $S2_{Mo}$-$MoS_2$-5[18,21]. Additionally, the Raman and PL spectra show that $S2_{Mo}$-$MoS_2$-5 has

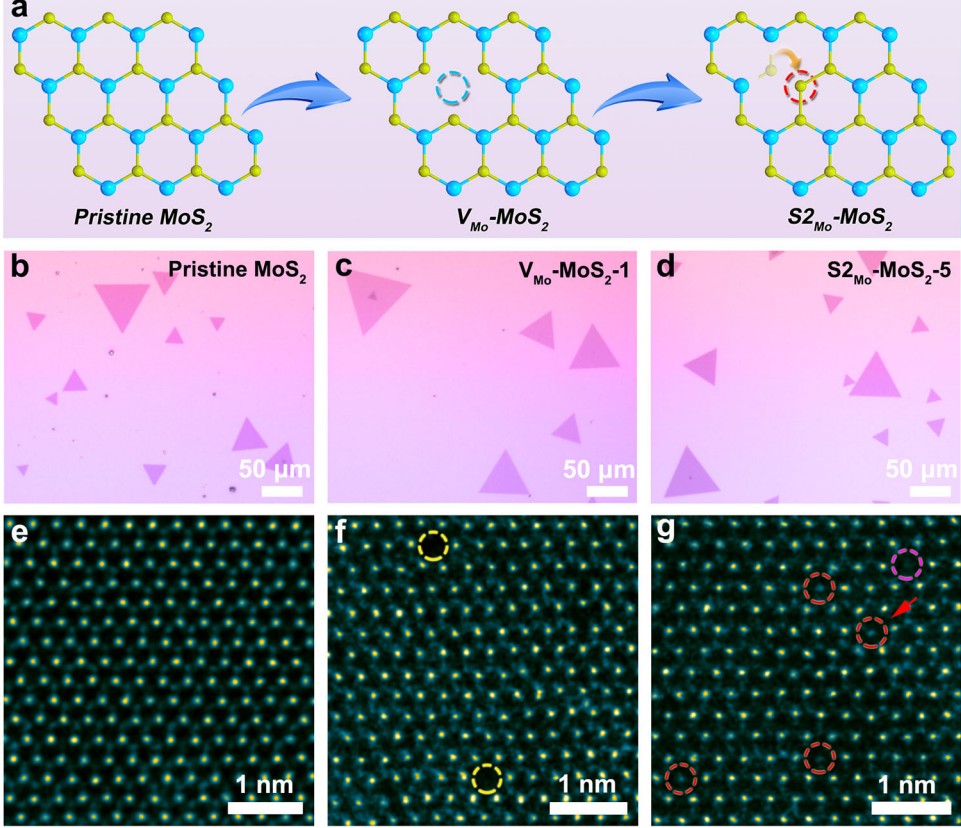

**Fig. 1 | Structural characterization of three $MoS_2$-based materials. a** Schematic diagram of the formation process of the antisite defect. **b**–**d** Typical optical images of the monolayer pristine $MoS_2$ (**b**), $V_{Mo}$-$MoS_2$-1 (**c**) and $S2_{Mo}$-$MoS_2$-5 (**d**). **e**–**g** Atomic-resolution HAADF-STEM images of the pristine $MoS_2$ (**e**), $V_{Mo}$-$MoS_2$-1 (**f**) and $S2_{Mo}$-$MoS_2$-5 (**g**).

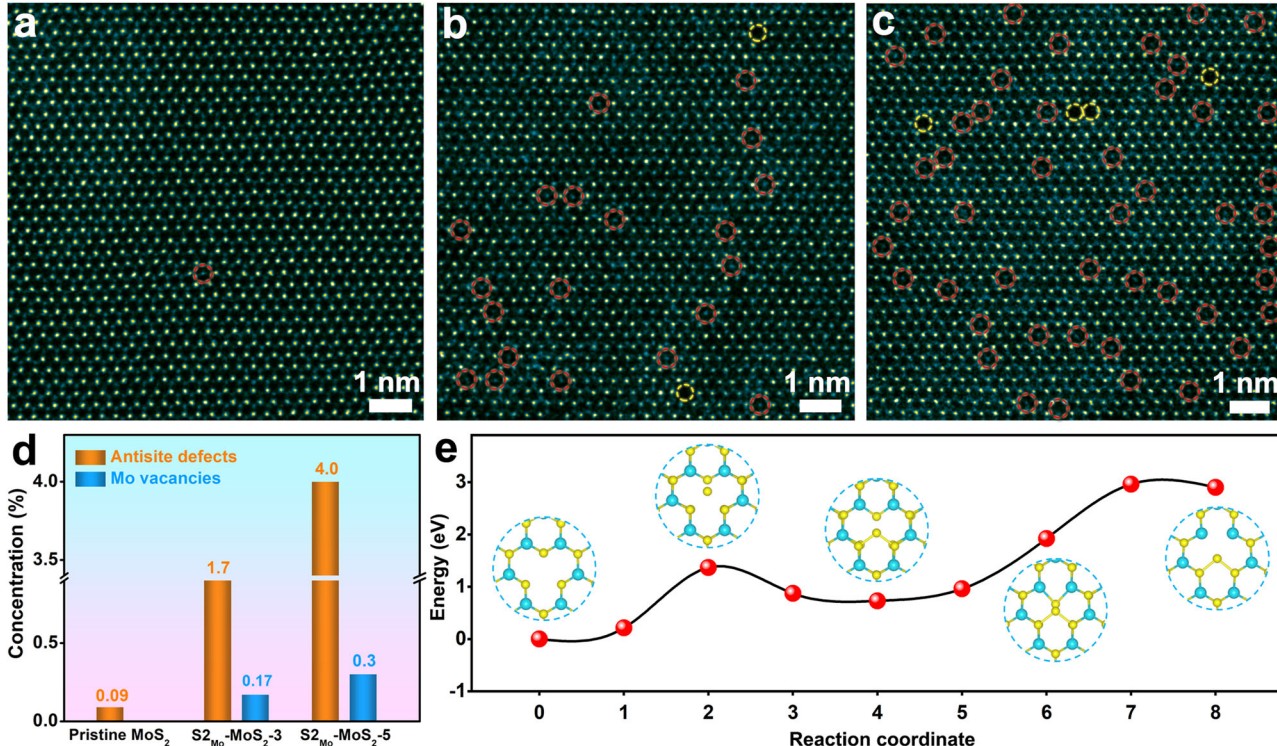

**Fig. 2 | Characterization of the atomic structure of the antisite defects at different concentrations and the calculation of the formation energy. a–c** Large-region atomic-resolution HAADF-STEM images of the pristine $MoS_2$ (**a**), $S2_{Mo}$-$MoS_2$-3 (**b**), and $S2_{Mo}$-$MoS_2$-5 (**c**). **d** Defect concentration statistics for the pristine $MoS_2$, $S2_{Mo}$-$MoS_2$-3, and $S2_{Mo}$-$MoS_2$-5. **e** Reaction path diagram for the step-by-step growth mechanism of the antisite defects in $S2_{Mo}$-$MoS_2$.

more defects than $V_{Mo}$-$MoS_2$, which may be due to the longer calcination time of $S2_{Mo}$-$MoS_2$-5.

Aberration-corrected scanning transmission electron microscopy (AC-STEM) was performed to reveal the atomic configurations of the three $MoS_2$-based materials. Firstly, there is no significant damage from the electron beam when collecting high-angle annular dark field (HAADF) images (Supplementary Fig. 4), which proves that the atomic structure of the prepared monolayer $MoS_2$ was relatively stable. Then, Fig. 1e shows the HAADF-STEM image of the pristine $MoS_2$, where high- and low-contrast atoms correspond to Mo and S, respectively. As shown in Fig. 1f, Mo vacancies (indicated by the yellow circles) were found in $V_{Mo}$-$MoS_2$-1, indicating that the Mo atoms first break away from their original sites to form vacancies during the annealing process. Further, the concentration of the Mo vacancies was calculated to be approximately 2.5 % through the larger-area atomic-resolution HAADF image of $V_{Mo}$-$MoS_2$-1 (Supplementary Fig. 5). The HAADF-STEM image of $S2_{Mo}$-$MoS_2$-5 is shown in Fig. 1g, and the antisite defects are represented by the red circles. We can clearly observe that, in each antisite defect, two S atoms occupy a Mo site ($S2_{Mo}$), where the two S atoms migrate from the nearby locations to the Mo site and leave the S vacancy, as indicated by the red arrow. Notably, the antisite defect type of a single S atom occupying a Mo site ($S_{Mo}$) has also been found, as indicated by the magenta arrow in Fig. 1g. Accordingly, more data on the HAADF-STEM atomic intensity analysis of the antisite defects are shown in Supplementary Fig. 6. These results further confirm the successful preparation of the antisite defects.

Then, we explore the effect of the antisite defects concentration under the same experimental conditions except for different annealing times. It is worth mentioning that without annealing treatment, one can hardly find antisite defects with extremely low concentrations in the pristine monolayer $MoS_2$ (Fig. 2a), not to mention establishing the qualitative or quantitative relationship between such defects and any resulting catalytic properties. Further, more HAADF-STEM data show

that there are no S vacancies around the antisite defects in pristine $MoS_2$ (Supplementary Fig. 7), indicating that foreign S atoms can directly occupy Mo sites to form the antisite defects. This is also consistent with the atomic configuration of the antisite defects reported in the literature[22,23]. When the annealing time is 3 min for pristine $MoS_2$ (denoted as $S2_{Mo}$-$MoS_2$-3), we also find the existence of Mo vacancies in addition to the antisite defects (Fig. 2b), indicating that Mo vacancies may be the prerequisite for the formation of the antisite defects. Besides, in the $S2_{Mo}$-$MoS_2$-5 samples, the concentration of the antisite defects is approximately 4.0% (Fig. 2c). Compared with pristine $MoS_2$, we can control the concentration of the antisite defects in $S2_{Mo}$-$MoS_2$-5 to increase by 44 times, and the concentration of Mo vacancies also increases accordingly (Fig. 2d). As the annealing time increases to 15 min (denoted as $S2_{Mo}$-$MoS_2$-15), antisite defects can still be found; however, more hole structures also appear (Supplementary Fig. 8), indicating that the monolayer $MoS_2$ material has been destroyed.

In addition, in the $S2_{Mo}$-$MoS_2$-5 sample, we also find that different S vacancies may appear around antisite defects (Supplementary Fig. 9). DFT calculations (Supplementary Fig. 10) further demonstrate that S vacancies could be beneficial for the formation of Mo vacancies and the formation energy gradually decreases with increasing S vacancies. In fact, the formation of Mo vacancies is a critical prerequisite for the generation of the $S2_{Mo}$ defects, which vacate position space for the antisite S atoms. Furthermore, the climbing image nudged elastic band (cNEB) method was performed to understand the underlying growth mechanism of the antisite defects. Considering that there may be multiple growth pathways to form the antisite defects during the annealing process, we compared and analyzed the diffusion barriers of several growth pathways to determine the most likely ones. First, Supplementary Fig. 11a shows the energy profile for the simultaneous migration of two S atoms directly to occupy a Mo vacancy site to form an $S2_{Mo}$ antisite defect. The energy barrier of 3.1 eV indicates

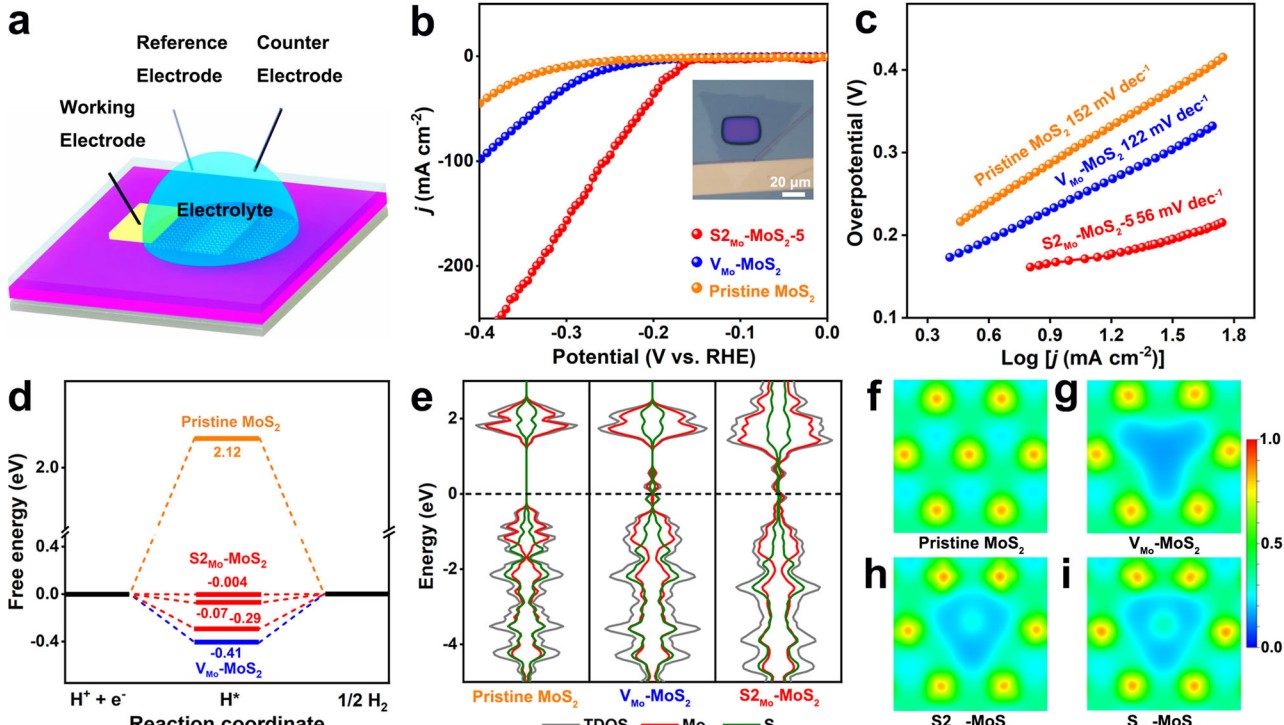

**Fig. 3 | HER performance evaluation and DFT calculation of pristine MoS₂, V_{Mo}-MoS₂, and S2_{Mo}-MoS₂. a** Schematic diagram of a micro-electrochemical device. **b** Polarization curves and (**c**) Tafel plots of pristine MoS₂, V_{Mo}-MoS₂, and S2_{Mo}-MoS₂. The inset in **b** is the optical image of a micro-electrochemical device. **d** HER free-energy diagram with detailed $\Delta G_{H^*}$ values for pristine MoS₂, V_{Mo}-MoS₂ and S2_{Mo}-MoS₂. **e** Total and partial density of states (DOS) of pristine MoS₂, V_{Mo}-MoS₂, and S2_{Mo}-MoS₂. The Fermi level is set to zero and denoted by the black dotted line. **f–i** Charge density distributions for pristine MoS₂, V_{Mo}-MoS₂, S2_{Mo}-MoS₂, and S_{Mo}-MoS₂. By definition, a region with a value of 1.0 denotes ideal charge accumulation, whereas a region with a value near 0.0 signifies a remarkably low charge density.

that this direct formation process is relatively difficult. Figure 2e displays the energy profile for another step-by-step growth mechanism, in which one S atom first migrates and leaves another S atom in its original position to form an intermediate S_{Mo} defect, then the other one further migrates to form the S2_{Mo} defect. The energy barrier for forming the S_{Mo} defect is ~1.36 eV, indicating that it is relatively easy to form S_{Mo} defects, which has been clearly observed in the experiment shown in Fig. 1g and Supplementary Fig. 6. It is worth noting that compared to Supplementary Fig. 11a, the growth process of forming S2_{Mo} defects from already existing S_{Mo} defects has a lower energy barrier of ~2.23 eV, which suggests that such a step-by-step growth pathway is kinetically more favorable. Therefore, the intermediate S_{Mo} antisite defects play a crucial bridging role in the formation of the S2_{Mo} antisite defects. Further, other growth mechanisms of the S2_{Mo} antisite defects have also been reasonably speculated (Supplementary Figs. 11 and 12).

## Electrocatalytic HER performance and DFT calculations
To explore the effect of different electric field distributions in monolayer MoS₂ on its catalytic performance, a homemade micro-electrochemical HER test device is used to accurately detect the influence of the MoS₂ base plane catalytic performance by the defect structure. As shown in Fig. 3a, a micro-electrochemical catalytic device at an exact location on the MoS₂ nanosheets base plane is constructed, and it can avoid the interference of numerous external factors and the edge unsaturated active sites[18,24]. Meanwhile, the window exposed region in the electrochemical device is in close contact with the 0.5 M H₂SO₄ electrolyte, forming an effective and accurate electrochemically active surface area (ECSA)[18]. The influence of the number of active sites per unit area on the HER performance of the 2D MoS₂ nanosheet can be effectively evaluated by using the ESCA (including the determined number of active sites for all 2D MoS₂ samples). Therefore, based on

ECSA of the micro-electrochemical catalytic devices, the polarization curves of each sample (pristine MoS₂, V_{Mo}-MoS₂-1 and S2_{Mo}-MoS₂-5) are accurately analyzed in Fig. 3b. At the current density of 10 mA cm⁻², S2_{Mo}-MoS₂-5 with an antisite defect structure exhibits a lower overpotential (169 mV) than those of pristine MoS₂ (303 mV) and V_{Mo}-MoS₂−1 (243 mV), indicating that the S2_{Mo}-MoS₂-5 sample needs a smaller overpotential to drive HER. Meanwhile, the Tafel plots of the three MoS₂-based samples were evaluated from the polarization curves in Fig. 3c. S2_{Mo}-MoS₂-5 gives the lowest Tafel slope of 56 mV dec⁻¹ compared to pristine MoS₂ and V_{Mo}-MoS₂-1. Furthermore, multiple S2_{Mo}-MoS₂-5 samples were conducted to electrocatalytic HER tests (Supplementary Fig. 13). The results show that the HER performance remained relatively stable, which illustrates the rough homogeneity of the number of antisite defects in S2_{Mo}-MoS₂-5 samples. Hence, all these indicators imply that the antisite defect is an excellent catalytically active site for HER.

Then, DFT calculations were performed to reveal the mechanism of enhanced catalytic activity of the antisite defects. The Gibbs free energy of H* adsorption could be used to evaluate HER catalytic activity by plotting the free energy diagram of $\Delta G_{H^*}$. Figure 3d summarizes the calculated $\Delta G_{H^*}$ at different active sites on pristine MoS₂, V_{Mo}-MoS₂, and S2_{Mo}-MoS₂. Given the actual existence of antisite defect structures with different S vacancies in the experiment in Supplementary Fig. 9, we also constructed several vacancy configurations and investigated their catalytic activities. The detailed atomic structures and active sites are shown in Supplementary Fig. 14. The HER free-energy diagrams in Fig. 3d indicate that the S2_{Mo}-MoS₂ structure could optimize the adsorption of H* over a large range and several $\Delta G_{H^*}$ values are superior to those of MoS₂ and V_{Mo}-MoS₂, thus greatly improving the HER activity. This result further shows that the optimal reaction centers mainly originate from the antisite defect structures with S vacancies and surrounding S atoms as active sites in

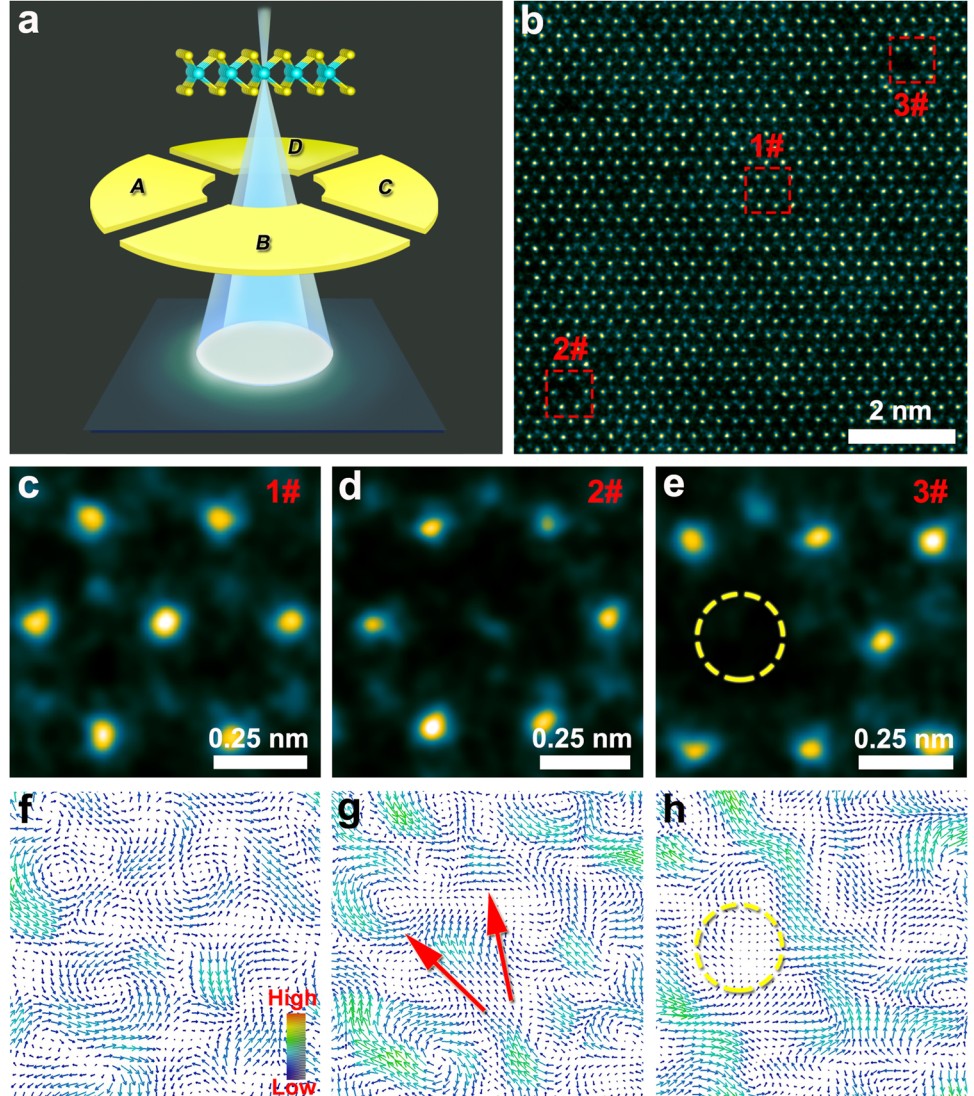

**Fig. 4 | DPC characterization results of monolayer S2$_{Mo}$-MoS$_2$-5. a** Imaging mechanism of DPC-STEM. **b** HAADF-STEM imaging area of S2$_{Mo}$-MoS$_2$-5 corresponding to the electric field signal collected by the DPC segment detector. **c–e** Enlarged regions of pristine atomic structure (**c**), the antisite defect structure (**d**), and Mo vacancy structure (**e**), which correspond to the regions of 1#, 2#, and 3# in (**b**). **f–h** Atomic electric field distribution images corresponding to (**c, d**). The directions of the arrows in (f·h) indicate the directions of the electric fields, and the different colors of the arrows indicate the intensities of the electric fields. The red arrows in (**g**) indicate the electric field polarization region.

Supplementary Fig. 14, demonstrating that the excellent HER catalytic performance of S2$_{Mo}$-MoS$_2$ depends on the co-existence of various antisite defect structures. Furthermore, the coordination bonding-based analysis is used to elucidate the variations in hydrogen adsorption energies. For pristine MoS$_2$, the nearly saturated bonding state of S atoms makes the adsorption of additional H* difficult (Supplementary Fig. 15a), resulting in weak hydrogen adsorption. For V$_{Mo}$-MoS$_2$ in Supplementary Fig. 15b, the introduction of V$_{mo}$ leads to two-coordinated S atoms with two Mo-S bonds, resulting in an unsaturated property that causes relatively stronger H* adsorption. In Supplementary Fig. 15c, d, unlike in V$_{Mo}$-MoS$_2$, the two-coordinated S atoms in S2$_{Mo}$-MoS$_2$ structures exhibit more pronounced structural distortions, which will weaken the adsorption of H* compared to the symmetric two-coordinated S atoms in V$_{Mo}$-MoS$_2$. Additionally, for the unique three-coordinated S atom with two Mo-S bonds and one S–S bond, the weaker S-S bond than the Mo–S bond allows for a greater redundancy of bonding energy to adsorb H* than pristine MoS$_2$, thereby enhancing hydrogen adsorption. In contrast, the additional S–S bond would result in much weaker adsorption compared to two-coordinated S

atoms in V$_{Mo}$-MoS$_2$. Overall, the formation of antisite defects breaks the original coordination situation of active S atoms and introduces new coordination environments more favorable for H* adsorption than pristine MoS$_2$ and V$_{Mo}$-MoS$_2$, thus improving the catalytic activity.

Electrical conductivity is another key factor for catalytic activity, and good electrical conductivity can guarantee efficient electron transfer during HER. Figure 3e shows the total and partial density of states (DOS) of pristine MoS$_2$, V$_{Mo}$-MoS$_2$, and S2$_{Mo}$-MoS$_2$. It can be clearly seen that pristine MoS$_2$ exhibits semiconductor properties with an obvious band gap near the Fermi level. However, the existence of Mo vacancies and the antisite defects introduces electronic states through the Fermi level and induces a semiconductor-to-metallic transition, indicating enhanced electrical conductivity. Further, the integral DOS is calculated by integrating the occupied and unoccupied electronic states around the Fermi level. This result also indicates that S2$_{Mo}$-MoS$_2$ has a better electrical conductivity (Supplementary Fig. 16). Structural distortion is often accompanied by the redistribution of charge density. As shown in Fig. 3f and Supplementary Fig. 17, pristine MoS$_2$ shows a periodically distributed charge density. However, the

introduction of Mo vacancies and the antisite defects leads to local lattice distortions and charge density redistributions, which will have a significant effect on catalytic activity. For $V_{Mo}$-$MoS_2$, only the charge of S atoms around Mo vacancies is modulated and exhibits a triple symmetric distribution (Fig. 3g). However, due to the diversity of antisite defect structures, $S2_{Mo}$-$MoS_2$ exhibits completely different and asymmetric charge density distribution (Fig. 3h, i and Supplementary Fig. 17). The asymmetric charge density of antisite defect atoms and surrounding S and Mo atoms have been effectively modulated within a wide range, which activates the activity of these atoms to a greater extent for the adsorption of H* (Fig. 3d), thus improving catalytic efficiency. Therefore, the asymmetric charge density endows $S2_{Mo}$-$MoS_2$ with better catalytic performance. Moreover, it has also been previously proven that the asymmetric charge distribution on the catalyst surface can improve catalytic activity[25], but the essential polarization has not been proposed. Besides, to elucidate the impact of the antisite defects on the surface potential, this distribution of surface electrostatic potential was computed to more intuitively investigate the influence of antisite defects on the surface electric field potential under experimental conditions (Supplementary Fig. 18). In comparison to the defect-free $MoS_2$ at a significant separation, there is a significant variation in the potential at defect sites. This observation aligns well with the theoretical proposition that regions with lower electrostatic potential are more prone to electron donation.

### Characterization of atomic electric field distribution

The asymmetric charge density distribution on the catalyst surface is directly related to its electric field, which is the decisive origin of catalytic performances[6,11–16,26]. Here, the DPC technology, a recent advancement in STEM imaging[27–30], was selected to characterize the electric field distributions of the $MoS_2$-based materials. Figure 4a depicts the imaging mechanism of DPC. When the electron beam is close to an atom, the charge and the resultant electric field of the atom cause the intensity distributions to become asymmetric. The difference between the intensities detected by the segments of A and C (or B and D) is the DPC signal, which can be used to obtain the distributions of the electric field according to the established center of mass (COM)-based atomic-resolution DPC methodology[27–30]. It should be noted that DPC and COM have been successfully used to visualize and analyze the electric fields of graphene defects, single Au atoms, and $MoS_2$[27,29,31–33]. Therefore, it is reasonable to use DPC and COM to visualize and analyze the electric fields of atomic defects on the monolayer $MoS_2$ materials. Figure 4b shows the HAADF-STEM image corresponding to the DPC signal acquisition area of $S2_{Mo}$-$MoS_2$-5 (more data about the DPC signal are provided in Supplementary Fig. 19). The enlarged regions of the pristine structure, an antisite defect, and an Mo vacancy are shown in Fig. 4c–e, respectively. The three regions are all from the same sample in Fig. 4b, eliminating the potential impact of different samples or external conditions on DPC results.

Figure 4f shows the periodic atomic electric field distribution corresponding to the region of the pristine $MoS_2$ structure in Fig. 4c. Further, the electric field distribution corresponding to the atomic structure of the antisite defect in the region of Fig. 4d is shown in Fig. 4g. Notably, we find that the asymmetric electric field distribution of the S atom occupying the Mo site is polarized, and the polarization direction is shown by the red arrow in Fig. 4g, which is significantly different from the electric field distribution of the original Mo atom (data of more polarizations in atomic electric fields are shown in Supplementary Fig. 20). Besides, the simulation image results also show that the electric field distribution of the antisite defect structure has obvious asymmetric polarization (Supplementary Fig. 21). Then, Fig. 4h shows the electric field distribution (the yellow circle) of the Mo vacancy corresponding to Fig. 4e. The electric field distribution signal at the vacancy is weakened due to the lack of atoms, resulting in

disturbances on the electric field distribution of the atoms around the vacancy. In addition, overlay images of HAADF-STEM and the corresponding DPC map in Fig. 4c-h more clearly show the electric field distribution of individual atoms in monolayer $MoS_2$ (Supplementary Fig. 22). In brief, the electric field distribution of different atomic structures of monolayer $MoS_2$ is well demonstrated by DPC. Further, we obtained a differentiated DPC (dDPC) mapping image of the antisite defect (Supplementary Fig. 23), and the result shows that the charge density distribution of the antisite defect structure was asymmetric, which is also consistent with the DFT calculations. Combining the DPC, dDPC and the DFT calculations results indicate that the polarization in the atomic electric fields of antisite defects directly leads to the appearance of asymmetric charge distributions, enhancing the adsorption of H* and further optimizing the HER catalytic activity.

## Discussion

In summary, we constructed antisite defects on the surface of monolayer $MoS_2$, and the atomic formation process of the antisite defects is disclosed via AC-STEM and DFT calculations. Then, accurate testing with micro-electrochemical devices elucidated the significant improvement of the antisite defects on the HER activity of monolayer $MoS_2$. Further, the DPC technology revealed the polarization in the atomic electric field distribution of the antisite defects. Together with DFT, our results indicate that the asymmetric charge distribution caused by the polarized electric field of the antisite defects can promote the adsorption of H*. This is the first example to explore the relationship between the atomic-level polarization in electric fields of catalyst surfaces and their catalytic activity. This study not only correlates the influence of atomic-level defects on catalytic activity but also paves the way to potential real-time catalysis studies using micro-electro-mechanical systems (MEMS)-based devices in microscopes.

## Methods

### Chemicals

The $SiO_2$/Si (270 nm $SiO_2$) substrates were purchased from Suzhou Ruicai Semiconductor Co., Ltd.; $MoO_3$ (≥99.5%) and S powder (≥99.5%) were purchased from Sigma-Aldrich (Shanghai) Trading Co., Ltd.; NaCl (≥99.5%) were purchased from Shanghai Aladdin Biochemical Technology Co., Ltd.

### Synthesis of monolayer $V_{Mo}$-$MoS_2$ and $S2_{Mo}$-$MoS_2$

Monolayer $MoS_2$ was grown as we reported before[18,34]. The grown pure $MoS_2$ was placed in the middle of a new quartz tube. Later, Ar (300 sccm) was used to ventilate for 10 min and remove the impurities in the tube of the system. $Ar/H_2$ (95 sccm and 5 sccm) gas mixture was maintained to flow in the system for annealing treatment. Then, the furnace was heated to the specified temperature (400 °C) within 60 min, and kept at 400 °C for 1 min to 5 min. Finally, the CVD furnace was fleetly cooled to room temperature. After that, 2D $V_{Mo}$-$MoS_2$ and $S2_{Mo}$-$MoS_2$ nanosheets were then manufactured.

### Structural characterization

Optical microscopy (Nikon H600L) and atomic force microscope (AFM, Bruker Dimension Icon) were used to observe the morphologies of these 2D samples. Horiba Instruments INC (1024×256-OE) equipped with a 532 nm laser excitation was used to represent the Raman and PL spectrum. The AC-STEM characterization was performed using a ThermoFisher Themis Z microscope equipped with two aberration correctors under 300 kV and 15 pA.

### DPC characterizations

According to the papers of Chapman et al.[35] and Rose[36] and the recently published ones about obtaining atomic electric fields in

atomic-resolution imaging[27,29], we use the following formula.

$$I_{COM,i}(R) = \sum_j \{k_i\}_{COM,j} I_j(R)$$

(1)

in which $\{k_i\}_{COM,j}$ is the $x$- or $y$-directional component of the COM of a DPC detector segment numbered as $j$. That is, $i = x$ or $y$. The values of $\{k_i\}_{COM,j}$ are determined by the geometry of the DPC detector. $I_j(R)$ is the normalized electron intensity detected by the $j$ detector segment at a position with the coordinate of $R$ in the measured specimen. The values of $I_j(R)$ and $R$ can be found in the corresponding DPC-STEM images. When all $I_{COM,i}(R)$ values calculated by the formula are plotted versus their corresponding $R$ values, a 2D vector COM map is obtained. The software used to perform the calculations of Formula is DigitalMicrograph, and the one used to draw the vector maps is Avizo. DPC-STEM image simulation was performed using the Dr. Probe V1.10 software package[37].

## Device fabrication
According to our previous device fabrication method[18,24], these 2D samples on the SiO$_2$/Si substrate were spin-coated with SPR-220-3a photoresist at 4000 rpm for 1 min and then baked at 115 °C for 90 s. Next, an ultraviolet lithography machine was used to pattern the electrode pattern. Thermal evaporation was used to deposit with In/Au (10 nm/50 nm) to connect the monolayer 2D samples. The residual photoresist was removed by acetone following a lift-off process, and the metallic electrode was obtained for working electrode devices. In the micro-electrocatalytic process, the photoresist was spin-coated again on the devices in SiO$_2$/Si substrates, and baked at 115 °C for 90 s. Then, the window exposed region of these 2D samples was opened by a proper ultraviolet beam (the actual surface area of the exposed 2D materials) for HER measurements.

## HER measurements
The micro-electrocatalysis performance was tested by a three-electrode system using a CHI 760E electrochemical workstation. A graphite carbon electrode with a diameter of 1 mm tip served as the count electrode, and a homemade saturated Ag/AgCl electrode served as the reference electrode. Each measured monolayer 2D sample with the window-exposed region served as the working electrode. Each electrode is fixed accurately with a hanging beam arm. The HER activity was evaluated in 0.5 M H$_2$SO$_4$ electrolyte by linear sweep voltammetry at a scan rate of 5 mV s$^{-1}$. The electrolyte volume on the window surface of the 2D material was fixed at 5 μL. All reported potentials were converted to reversible hydrogen electrode (RHE) potentials, and no iR-corrected was used in all the electrochemical measurements.

## DFT calculations
The spin-polarized density functional theory (DFT) calculations are performed by using the Vienna Ab initio Simulation Package (VASP)[38,39]. The electronic exchange-correlation energy is described by the Perdew–Burke–Ernzerhof (PBE) functional within the generalized gradient approximation[40]. The projector augmented wave pseudopotential is used with a cutoff energy of 450 eV. All atoms are fully relaxed until the force acting on each atom is less than 0.02 eV/Å. For all calculations, a $5 \times 5 \times 1$ supercell is employed with a vacuum region of 15 Å to prevent negligible interlayer interactions between the MoS$_2$ and its mirror images. The Brillouin zone is sampled by a $4 \times 4 \times 1$ Monkhorst–Pack scheme k-point mesh. The climbing image nudged elastic band (cNEB) method[41] was performed to determine the growth pathway with minimum diffusion barrier for antisite defects. As in the structural optimization above, the same cut-off energies of 450 eV, the k-point mesh of $4 \times 4 \times 1$, and convergence criteria are used for the supercell with $5 \times 5 \times 1$ surface periodicity during the cNEB

calculations. Between the stable initial and final states, we inserted three points to determine the optimal growth paths.

The formation energy ($E_f$) of Mo vacancy ($V_{Mo}$) is calculated by:

$$E_f = E_{v_{Mo}-MoS_2} - E_{MoS_2} + \mu_{Mo}$$

(2)

Where $E_{MoS_2}$ and $E_{V_{Mo}-MoS_2}$ are the total energies of the MoS$_2$ surface without and with Mo vacancies, respectively. $\mu_{Mo}$ is the chemical potential of the Mo atom and is calculated from the bulk phase of Mo.

The Gibbs free energy of hydrogen adsorption ($\Delta G_{H^*}$) can be calculated by[42,43]:

$$\Delta G_{H^*} = \Delta E_{H^*} + \Delta E_{ZPE} - T\Delta S$$

(3)

where $\Delta E_{ZPE}$ and $\Delta S$ are the zero-point energy correction and the vibrational entropy change, respectively. $\Delta E_{H^*}$ is the hydrogen adsorption energy and is calculated by:

$$\Delta E_{H^*} = E_{H^*} - E_{surface} - 1/2 E_{H_2}$$

(4)

where $E_{H^*}$ and $E_{surface}$ are the total energies of the surface model with H* adsorption and the clear surface model, respectively. $E_{H_2}$ is the energy of the gas phase H$_2$ molecule.

An implicit solvent model was employed. The linearized Poisson–Boltzmann model with a Debye length of 3.0 Å mimics the compensating charge. The electrode potential of the two models to 0.2 V vs SHE was adjusted in accordance with experimental conditions, following the procedure outlined in Equation. The solvent environment was modeled by the VASPsol code[44–47].

$$U_q(V/SHE) = -4.6V - \Phi_q/e$$

(5)

where −4.6 V is the absolute electrode potential of the SHE benchmarked in VASPsol, and $-\Phi_q$ is the work function of the charged system.

## Data availability
All data supporting the findings of this study are available from the Source Data. Additional data are available from the corresponding authors upon reasonable request. Source data are provided in this paper.

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

## Acknowledgements

We acknowledge the financial support by the National Key R&D Program of China (2021YFA1202300), National Natural Science Foundation of China (52202373, 51971157, 22205209, 12104385), Natural Science Foundation of Jiangsu Province of China (BK20210729), Shenzhen Science and Technology Program (JCYJ20210324115412035, JCYJ20210324123202008, JCYJ20210324122803009, and ZDSYS20210813095534001), Guangdong Basic and Applied Basic Research Foundation (2021A1515110880), Henan Province Key Research and Promotion Project-Scientific and Technological Breakthroughs (232102230088). The computational resources were provided by the supercomputer TianHe-1 in Changsha, China.

## Author contributions

J.X., J.Luo, Y.Y., and J.Lu conceived and designed the project. G.S. performed the synthetic and electrochemical experiments. J.X. performed the electron microscopy experiment, to which C.J., K.H., and S.D. contributed. X.X. performed DFT calculations. P.J. performed image simulation. J.X. and G.S. analyzed the experimental data. J.X., G.S., J.Luo, Y.Y., and J.Lu wrote the manuscript. S.W., Y.Y., J.Luo, and J.Lu supervised the project. All authors discussed the results.

## Competing interests

The authors declare no competing interests.
