## [Peer review file · Nature Communications]

REVIEWER COMMENTS

Reviewer #1 (Remarks to the Author):

Jie Xu et al report the correlation of the structure and performance of monolayer MoS₂ for the hydrogen evolution reaction by tuning the atomic defects. With AC-STEM imaging and DPC-STEM technique, they have firstly determined the polarized electric field of antisite defect for facilitating the catalytic activity in this reaction. I recommend to publish this work in Nature Communications, however, the observation of DPC-STEM here need be addressed and improved with following concerns.

1. The voltage for the AC-STEM measurement in this work is 300 kV, which differs to the typical voltage of 60/80 kV used for the monolayer MoS₂ in most of studies as the high voltage in STEM could easily cause the radiation (knock-on) damage for this material. Therefore, the authors should explain if the original state of investigate materials is reversed in the observation of STEM and DPC under such condition or not.

2. There is no any simulation to support the polarized electric field distribution of the antisite defect observed by DPC in the experiment. The determination on the polarized in atomic field distribution images also should be addressed more clearly in Fig. 4.

3. Compared to the atomic electric field, the charge density mapping in atomic-scale is a more direct indicator for the asymmetric charge distribution of antisite defect. This could also be obtained by the DPC technique but missed in this work.

Reviewer #2 (Remarks to the Author):

I would like to complement the efforts put in by the authors in bringing out this very nice piece of challenging work. The importance of this study is showing the potential of differential phase contrast STEM imaging in revealing the atomic level surface charges which have an important consequence on several electrochemical phenomena. While the idea and the execution of the study supported by relevant theoretical modeling bring out the central idea proposed here, I have a few suggestions on the way the sentences are currently constructed as well as a few technical suggestions which in my opinion would make the claims in this study stronger.

Technical inputs:

1. The authors start by claiming in the abstract "For the first time, it has been experimentally proven that atomic-level polarization in electric fields can enhance catalytic HER activity", further in the text, they cite literature with the statement "A large number of advanced electrocatalysts have atomic defect structures on their surfaces. These defects can alter the electric field/charge distribution of electrocatalysts with enhanced catalytic performances⁷⁻¹³"

This is very correct and acceptable. Defects certainly influence the surface-potential/EDL. A very recently published article Xu, Y., Ma, Y. B., Gu, F., Yang, S. S., & Tian, C. S. (2023). Structure evolution at the gate-tunable suspended graphene-water interface. *Nature*, 1-5 also establishes something in these lines. From the DFT calculations done by the authors the link between the formation of defects and variation in surface potentials is shown whereas to show how this influences the catalytic activity, is it possible to establish something in the lines of this study of Swift, M. W., Swift, J. W., & Qi, Y. (2021). Modeling the electrical double layer at solid-state electrochemical interfaces. *Nature Computational Science*, 1(3), 212-220. If not a complete study, a 1-dimensional variation of potential across the cross-section of electrolyte-MoS₂ with the presence of a defect perhaps supports the claimed results substantially? From the previously published studies of the same group, it appears like they have ample experience in this methodology and computation to go one step further and show the variation of surface potential across a monolayer-electrolyte interface in the

presence and absence of defects.

2. On a monolayer like MoS₂, while imaging at 300kV, it is highly likely that you are creating defects while imaging. Do the authors observe this when they obtain a series of images from the same region?

3. If newer defects are formed potentially by the knock-on damage by the electron beam, you are constantly changing the surface potential by creating newer defects. Have any dose optimisation tests been done?

4. The creation of electrodes to carry out 3-electrode electrochemical measurements is certainly a commendable job. The device fabrication involves spincoating and liftoff. Doesn't the fabrication process alter the surface potentials/state of defects in the MoS₂. Wouldn't the creation of electrodes first on the substrate and transferring the MoS₂ later by a transfer method (Schneider, Grégory F., et al. "Wedging transfer of nanostructures." *Nano letters* 10.5 (2010): 1912-1916) give a more accurate estimation of surface potentials in the electrochemical measurements?

5. How reproducible are the electrochemical measurements? Are the Tafel slopes indicating a direct correlation with the defects based on the synthesis procedure, or is there an influence of the device fabrication method of spin-coating and lift-off and subsequent processes?

6. Overall, the authors observe a variation in the surface potential due to the presence of defects clearly from electrochemical measurements. They also simulate the same by introducing defects. If they manage to make it a little more clearer in the written text or support with an additional simulation that the surface potential may alter in presence of electrolyte and in addition introducing defects alters further and if they manage to establish a link with the electrochemical experimental observation, this would make the outcome even more stronger. Nonetheless, not having this strictly doesn't demerit any of the observations in this study.

Suggestions on rephrasing the sentences:

If I understand what the authors are trying to communicate, my advice is to simply a few of these sentences. Apart from these modifications, there are several typographical errors and misplacing articles which may be checked at the end.

1. Please consider rewriting the first few sentences in the abstract as: "The thriving field of atomic defect engineering towards advanced electrocatalysis relies on the critical role of electric field polarization at the atomic scale. While this is proposed theoretically, the spatial configuration, orientation, and correlation with specific catalytic properties of materials are yet to be understood." Instead of the present text

2. Please rephrase the following sentence: "As such, analysing the non-periodic..." as: "Analyzing the non-periodic electric field on the electrocatalyst with high spatial accuracy is crucial for understanding the catalytic mechanism. Unfortunately, since the technical challenges of atomic imaging currently hinder the characterization of the non-periodic electric fields surrounding specific atomic defects, understanding such microscopic mechanisms largely relies on theoretical calculations¹⁴⁻¹⁶."

3. The sentence "Therefore, MoS₂ with point defects..." may be simplified as: "Therefore, MoS₂, with point defects as a catalyst, is an ideal material system to explore the effect of electric field polarization of atomic defect sites on the as-tuned catalytic property and performance."

4. The sentence "In fact, the asymmetric charge density distribution on the catalyst surface is directly related to its electric field, which is the decisive origin of catalytic performances^{6,11-16,26}. Here, the DPC technology, a recent advancement in STEM imaging²⁷⁻³⁰, was selected to characterize the electric field distributions of the MoS₂-based materials."

5. With the experimental results presented in this study, the authors may go as far as making a stronger conclusion by modifying the last sentence in conclusions as: "This is the first example to explore the relationship between the atomic-level polarization in electric fields of catalyst surfaces and their catalytic activity. This study not only correlates the influence of atomic-level defects on catalytic activity but also paves the way to potential real-time catalysis studies using MEMS-based devices in microscopes."

Reviewer #4 (Remarks to the Author):

In this study, Xu et al. have undertaken a comprehensive investigation of electric field polarization and its impact on the hydrogen evolution reaction (HER) efficiency in defective MoS₂. Their work combines experimental techniques and theoretical calculations to shed light on the factors influencing catalytic activity. While the experimental section is commendable, there are some suggestions for refining the theoretical portion of the study before it can be considered for acceptance.

1. Figure 3d should include labels for the hydrogen adsorption free energies, even though this information is available in the supplementary material. This will enhance the clarity of the figure and make it easier for readers to interpret.
2. It is better to provide a quantitative explanation for the observed hydrogen adsorption energies in the context of electronic structure, possibly using theories such as band center theory. This would help readers better understand the underlying mechanisms driving the variations in adsorption energies for VMo-MoS₂, S₂Mo-MoS₂, and MoS₂.
3. To improve the clarity of the charge density diagrams, it is recommended to include a color bar that demonstrates charge accumulation and depletion. This will make it easier for readers to interpret the diagrams and understand the spatial distribution of charges.
4. In the Methods section, it is imperative to explicitly mention the use of the computational hydrogen electrode (CHE) method for calculating Gibbs free energy instead of simply using 1/2 H₂ directly. Additionally, cite the relevant literature, such as J. Phys. Chem. B 2004, 108(46), 17886–17892 and J. Electrochem. Soc. 2005, 152 (3), J23-J26. by J. K. Nørskov, to provide a reference for this approach, enhancing the transparency of the computational methodology.
5. Listing the detailed calculation parameters in the climbing image nudged elastic band (cNEB) method would be better.
6. Did the authors include the Van der Waals interactions and solvation correction in the DFT calculations?

Overall, these revisions will strengthen the theoretical aspect of the paper and improve its accessibility to the scientific community.

Responses to the Comments

Responses to the comments of Reviewer #1:

Comment:

Reviewer #1: Jie Xu et al report the correlation of the structure and performance of monolayer MoS₂ for the hydrogen evolution reaction by tuning the atomic defects. With AC-STEM imaging and DPC-STEM technique, they have firstly determined the polarized electric field of antisite defect for facilitating the catalytic activity in this reaction. I recommend to publish this work in Nature Communications, however, the observation of DPC-STEM here need be addressed and improved with following concerns.

1. The voltage for the AC-STEM measurement in this work is 300 kV, which differs to the typical voltage of 60/80 kV used for the monolayer MoS₂ in most of studies as the high voltage in STEM could easily cause the radiation (knock-on) damage for this material. Therefore, the authors should explain if the original state of investigate materials is reversed in the observation of STEM and DPC under such condition or not.

Response:

Thank you very much for reviewing our manuscript. We greatly appreciate your inspiring, useful, helpful, and constructive comments. Your concern regarding the damage to the monolayer MoS₂ structure caused by an operating voltage of 300 kV is quite reasonable. During the experimental process, we also carefully explored the impact of different voltages on HAADF imaging of monolayer pristine MoS₂. First, we used the ThermoFisher Themis Z microscope to characterize the atomic structure of monolayer MoS₂ under a voltage of 60 kV, as illustrated in **Figure R1**. Observing the S atom on the atomic resolution HAADF image in Figure R1 is very difficult, implying that the spatial resolution of images gathered under 60 kV conditions is inadequate, leading to certain atomic defect structures (such as antisite defects) going undiscovered. Therefore, we characterize the atomic electric field distribution of atomic defect structures using DPC technology to pursue an improved resolution.

Then, we found that at 300 kV, the atomic structure of monolayer MoS₂ and different defects can be clearly characterized, as shown in Figures 1e-g of the original version of the manuscript. Furthermore, we continuously acquired two HAADF images of the same local region of monolayer MoS₂ under 300 kV and 15 pA conditions, as shown in **Figure R2**. The results showed that the atomic structure of monolayer MoS₂ is relatively stable, and the electron beam causes no obvious damage when HAADF images are collected. Besides, some literature has also reported the use aberration-corrected scanning transmission electron microscopy (AC-STEM) for atomic resolution HAADF imaging of monolayer MoS₂ under 300 kV or 200 kV (*Nat. Commun.* 2022, 13, 3063; *J. Am. Chem. Soc.* 2023, 145, 11348), and the characterization results also indicate that electron beam will not cause obvious structural damage monolayer MoS₂. These results indicate that the monolayer MoS₂ prepared in this work possesses relatively stable structure under electron beam, and collecting their HAADF and DPC images under 300 kV conditions should not cause any observous structural damage and thus should not affect the reported results here.

The related results of **Figure R2** have been added as Supplementary Fig. 4 in Page 5 of the revised Supplementary Information (SI) file. For the revision, the following discussion is added on Pages 6 and 7 of the main text: *“Firstly, there is no significant damage from the electron beam when collecting high-angle annular dark field (HAADF) images (Supplementary Fig. 4), which proves that the atomic structure of the prepared monolayer MoS₂ was relatively stable”*. For your convenience, all changes have been highlighted by yellow in the revised manuscript and SI file.

Figure R1. (a,b) HAADF image and corresponding Wiener filter image of monolayer pristine MoS₂ under a voltage of 60 kV.

Figure R2. (a,b) Two HAADF images of monolayer pristine MoS₂ in the same micro region were collected continuously under 300 kV and 15 pA conditions. The results showed that the atomic structure of monolayer MoS₂ is not significantly changed under electron beam radiation when HAADF images are collected.

Comment:

2. There is no any simulation to support the polarized electric field distribution of the antisite defect observed by DPC in the experiment. The determination on the

polarized in atomic field distribution images also should be addressed more clearly in Fig. 4.

Response:

Thank you for this helpful suggestion. As you suggested, necessary simulations are important support for the polarized electric field distribution of the antisite defect results in experiments. Firstly, we performed DPC technology with four segments detector imaging simulation of electric field distribution using Dr. Probe V1.10 software package (Ultramicroscopy 2018, 193, 1). The model is a monolayer of MoS₂ with antisite defects structure with a thickness of 0.55 nm that was equidistantly divided into three layers for simulations based on multislice method. The simulations used a frozen lattice configuration of 100 variants per slice and were performed at 300 kV experimental conditions. To account for partial spatial coherence of the probes, the images were convolved with a probe size of 0.06 nm. Then, we also draw the vector maps with Avizo software, and the simulated electric field distribution of antisite defects structure are obtained in **Figure R3**. We also found significant asymmetric polarization in the electric field distribution of antisite defects structure from the simulation image (red arrows represent the electric field polarization region), further supporting the reliability of the DPC experimental results.

In addition, the display of the electric field distribution in Figures 4f-h of the original manuscript was not clear enough, so we redrawn the electric field distribution in Figures 4f-h to more clearly determine the differences in electric field distribution among different atomic structures. Moreover, we further overlay the electric field distribution on Figures 4f-h and HAADF images on Figures 4c-e, as shown in **Figure R4**, which can better display the atomic polarization electric field distribution at the antisite defect structure.

The related results of **Figures R2** and **R3** have been added as Supplementary Figs. 21, and 22 in Pages 22 and 23 of the revised SI file. On page 30 of the revised manuscript file, (Ultramicroscopy 2018, 193, 1) has been cited as Ref. 37. For the revision, the following discussion is added on Pages 18, and 21 of the main text: *“Besides, the simulation image results also show that the electric field distribution of the antisite defect structure has obvious asymmetric polarization (Supplementary Fig. 21)”*, *“In addition, overlay images of HAADF-STEM and corresponding DPC map in Fig.*

4c-h more clearly show the electric field distribution of individual atoms in monolayer MoS₂ (Supplementary Fig. 22)”, and “DPC-STEM image simulation was performed using the Dr. Probe V1.10 software package³⁷”.

Figure R3. (a,b) Simulated HAADF image and corresponding DPC image of monolayer MoS₂ with the antisite defect structure. The red arrows in (b) indicated the electric field polarization region.

Figure R4. (a-c) The overlay images of HAADF-STEM and corresponding DPC map of pristine atomic structure (a), the antisite defect structure (b) and Mo vacancy structure (c) in monolayer MoS₂.

Comment:

3. Compared to the atomic electric field, the charge density mapping in atomic-scale is a more direct indicator for the asymmetric charge distribution of antisite defect. This could also be obtained by the DPC technique but missed in this work.

Response:

Thank you very much for your constructive comment. DPC mapping with atomic resolution reflects the direction and intensity of electric fields distribution around individual atoms. In fact, we can further process DPC mapping to obtain its charge density distribution, which is called differentiated differential phase contrast (dDPC) mapping. Notably, dDPC mapping requires mathematical differentiation of the DPC signal, which can lead to a significant weakening of the dDPC signal strength, especially on charge density distribution of individual atoms. First of all, we further obtained dDPC mapping images in **Figure R5** of the corresponding regions of Figures 4b,d in the manuscript, and the results showed that the charge distribution of antisite defects structure was clearly asymmetric, which is different from the elliptical charge distribution of the surrounding Mo atom. This result is consistent with the theoretical calculation. However, we also found some abnormal distortions in the charge distribution intensity of other atoms, which is related to the surface structure in our synthesized materials. However, it is more likely that the current dDPC technology with four-segment detector does not have enough precision in characterizing the charge distribution of individual atoms.

Further, to verify the accuracy of dDPC technology in characterizing the charge density distribution of individual defect atoms, we prepared monolayer MoS₂ doped with Pt single atoms (namely Pt-MoS₂). From the atomic-resolution HAADF image in **Figure R6a**, it can be clearly seen that a single Pt atom occupies the Mo site. In theory, the charge density distribution of a heavy metal Pt single atom should be significantly different from that of the surrounding Mo atoms. However, from the atomic-resolution dDPC image obtained in the corresponding region of Figure R5a in **Figure R6b**, it is difficult to see an obvious difference in the charge density distribution of Pt single atoms compared to nearby Mo atoms. These results indicated that the accuracy of dDPC technology with four-segments detector may make it difficult to distinguish the difference in the charge density distribution of individual atoms.

The related results of **Figure R5c,d** have been added as Supplementary Fig. 23 in Page 24 of the revised SI file. For the revision, the following discussion is added on Page 18 of the main text: *“Further, we obtain differentiated differential phase contrast (dDPC) mapping image of the antisite defect (Supplementary Fig. 23), and the result*

shows that the charge density distribution of antisite defects structure was asymmetric, which is also consistent with the DFT calculations”.

Figure R5. (a,b) HAADF-STEM image and corresponding dDPC image of $S_{2Mo}-MoS_2-5$; (c,d) Enlarged regions of the positions indicated by the dashed lines in (a) and (b), the atomic structure of antisite defects and corresponding charge density distribution mapping.

Figure R6. (a,b) HAADF-STEM image and corresponding dDPC image of Pt-MoS₂. The red dashed circles in (a) represent the single Pt atom, and black dashed circles in (b) represent the charge density distribution of the single Pt atom.

Responses to the comments of Reviewer #2:

Comment:

Reviewer #2: I would like to complement the efforts put in by the authors in bringing out this very nice piece of challenging work. The importance of this study is showing the potential of differential phase contrast STEM imaging in revealing the atomic level surface charges which have an important consequence on several electrochemical phenomena. While the idea and the execution of the study supported by relevant theoretical modeling bring out the central idea proposed here, I have a few suggestions on the way the sentences are currently constructed as well as a few technical suggestions which in my opinion would make the claims in this study stronger.

Technical inputs:

1. The authors start by claiming in the abstract “For the first time, it has been experimentally proven that atomic-level polarization in electric fields can enhance catalytic HER activity”, further in the text, they cite literature with the statement “A large number of advanced electrocatalysts have atomic defect structures on their surfaces. These defects can alter the electric field/charge distribution of electrocatalysts with enhanced catalytic performances⁷⁻¹³”

This is very correct and acceptable. Defects certainly influence the surface-potential/EDL. A very recently published article Xu, Y., Ma, Y. B., Gu, F., Yang, S. S., & Tian, C. S. (2023). Structure evolution at the gate-tunable suspended graphene–water interface. *Nature*, 1-5 also establishes something in these lines. From the DFT calculations done by the authors the link between the formation of defects and variation in surface potentials is shown whereas to show how this influences the catalytic activity, is it possible to establish something in the lines of this study of Swift, M. W., Swift, J. W., & Qi, Y. (2021). Modeling the electrical double layer at solid-state electrochemical interfaces. *Nature Computational Science*, 1(3), 212-220. If not a complete study, a 1-dimensional variation of potential across the cross-section of electrolyte-MoS₂ with the presence of a defect perhaps supports the

claimed results substantially? From the previously published studies of the same group, it appears like they have ample experience in this methodology and computation to go one step further and show the variation of surface potential across a monolayer-electrolyte interface in the presence and absence of defects.

Response:

Thank you very much for reviewing our manuscript. We greatly appreciate your inspiring, useful, helpful and constructive comments. First of all, we read the literature you recommended (*Nature* 621, 506–510 (2023); *Nat. Comput. Sci.* 1, 212–220 (2021)) and consider the advice you suggested. To elucidate the impact of the antisite defects on the surface potential, we conducted calculations on two models with different antisite defects, represented in **Figure R1a** and **R1b**. Here the distribution of surface electrostatic potential was computed to more intuitively investigate the influence of defects on the surface electric field potential under experimental conditions. Additionally, considering computational efficiency, an implicit solvent model was employed. The linearized Poisson–Boltzmann model with a Debye length of 3.0 Å mimics the compensating charge.

We adjusted the electrode potential of the two models to 0.2 V vs SHE in accordance with the experimental conditions, following the procedure outlined in Equation. The solvent environment was modeled by the VASPsol code (*J. Chem. Phys.* **151**, 234101 (2019); *J. Chem. Phys.* **140**, 084106 (2014)).

$$U_q (\text{V/SHE}) = - 4.6 \text{ V} - \Phi_q / e$$

where - 4.6 V is the absolute electrode potential of the SHE benchmarked in VASPsol, and $-\Phi_q$ is the work function of the charged system.

Subsequently, under these conditions, we performed Electrostatic Potential (ESP) analyses on the models with two distinct defects, respectively. Multiwfn software was employed during the process of generating the plots. It is evident that, in comparison to the defect-free MoS₂ at a significant separation, there is a significant variation in the potential at defect sites. This observation aligns well with the theoretical proposition that regions with lower electrostatic potential are more prone to electron donation. This correlation aligns effectively with the adsorption energies previously calculated in our study.

Red represents a low electrostatic potential, indicating that this region more easily gives electrons and is more nucleophilic than other regions, while blue represents a high electrostatic potential, indicating that this region is easier to obtain electrons and is more electrophilic than other regions. Calculations use the 0.001 electrons/bohr³ density isosurface.

The related results of **Figure R1** have been added as Supplementary Fig. 18 in Page 19 of the revised Supplementary Information (SI) file. On page 31 of the revised manuscript file, [*Nature* 621, 506–510 (2023); *Nat. Comput. Sci.* 1, 212–220 (2021); *J. Chem. Phys.* **151**, 234101 (2019); *J. Chem. Phys.* **140**, 084106 (2014)] have been cited as Refs. 44-47. For the revision, the following discussion is added on Pages 15 and 24 of the main text: *“Besides, to elucidate the impact of the antisite defects on the surface potential, this distribution of surface electrostatic potential was computed to more intuitively investigate the influence of antisite defects on the surface electric field potential under experimental conditions (Supplementary Fig. 18). In comparison to the defect-free MoS₂ at a significant separation, there is a significant variation in the potential at defect sites. This observation aligns well with the theoretical proposition that regions with lower electrostatic potential are more prone to electron donation”, and “An implicit solvent model was employed. The linearized Poisson–Boltzmann model with a Debye length of 3.0 Å mimics the compensating charge. The electrode potential of the two models to 0.2 V vs SHE was adjusted in accordance with experimental conditions, following the procedure outlined in Equation. The solvent environment was modeled by the VASPsol code⁴⁴⁻⁴⁷.*

$$U_q (\text{V/SHE}) = - 4.6 \text{ V} - \Phi_q / e$$

where - 4.6 V is the absolute electrode potential of the SHE benchmarked in VASPsol, - Φ_q is the work function of the charged system.”

For your convenience, all changes have been highlighted by yellow in the revised manuscript and SI file.

Figure R1. (a,b) Mapping Electrostatic Potential on electron density isosurface of two kinds of antisite defects.

Comment:

2. On a monolayer like MoS₂, while imaging at 300 kV, it is highly likely that you are creating defects while imaging. Do the authors observe this when they obtain a series of images from the same region?

Response:

Thank you for your helpful and constructive comment. In fact, the higher the operating voltage of aberration-corrected scanning transmission electron microscopy (AC-STEM), the higher the spatial resolution of the collected image in theory. During the experimental process, we also carefully explored the impact of different voltages on HAADF imaging of monolayer MoS₂. Firstly, we used the AC-STEM to characterize the atomic structure of the monolayer MoS₂ under a voltage of 60 kV, as illustrated in **Figure R2**. Observing the S atom on the atomic resolution HAADF image in **Figure R2** is very difficult, implying that the spatial resolution of images gathered under 60 kV conditions is inadequate, leading to certain atomic defect structures (such as antisite defects) going undiscovered. Then, we found that under the condition of 300 kV, the atomic structure of monolayer MoS₂ and different defects can be clearly characterized by aberration-corrected scanning transmission electron microscopy (AC-STEM) as shown in Figures 1e-g of the original version manuscript. Furthermore, we continuously acquired two HAADF images of the same micro region of monolayer MoS₂ under 300 kV conditions in **Figure R3**. The results showed that the atomic

structure of monolayer MoS₂ is relatively stable, and the electron beam causes no obvious damage when HAADF images are collected. Besides, some literature has also reported the use AC-STEM for atomic resolution HAADF imaging of monolayer MoS₂ under 300 kV or 200 kV (*Nat. Commun.* 2022, 13, 3063; *J. Am. Chem. Soc.* 2023, 145, 11348), and the characterization results also indicate that electron beam will not cause monolayer MoS₂ structural damage. These results indicate that the monolayer MoS₂ atomic structure we prepared is relatively stable, and collecting their HAADF images under 300 kV conditions will not damage their atomic structure.

The related results of **Figure R2** have been added as Supplementary Fig. 4 in Page 5 of the revised SI file. For the revision, the following discussion is added on Pages 6 and 7 of the main text: *“Firstly, there is no significant damage from the electron beam when collecting high-angle annular dark field (HAADF) images (Supplementary Fig. 4), which proves that the atomic structure of the prepared monolayer MoS₂ was relatively stable”*.

Figure R2. (a,b) HAADF image and corresponding Wiener filter image of monolayer pristine MoS₂ under a voltage of 60 kV.

Figure R3. (a,b) Two HAADF images of monolayer MoS₂ in the same micro region were collected continuously under 300 kV conditions. The results showed that the atomic structure of monolayer MoS₂ is relatively stable under electron beam radiation when HAADF images are collected.

Comment:

3. If newer defects are formed potentially by the knock-on damage by the electron beam, you are constantly changing the surface potential by creating newer defects. Have any dose optimisation tests been done?

Response:

Thank you for your helpful and constructive comment. We obtained HAADF images of monolayer MoS₂ through AC-STEM at a voltage of 300 kV, and the dose of the electron beam is a crucial parameter. In order to verify the effect of electron beam dose on the HAADF imaging damage of the atomic structure of monolayer MoS₂, we explored the atomic-resolution HAADF imaging of monolayer MoS₂ under different dose conditions. On the same monolayer MoS₂ nanosheets, we collected atomic-resolution HAADF images of under different current conditions of 15 pA and 70 pA respectively, as shown in **Figure R4**. The results show that the atomic structure of monolayer MoS₂ has been the knock-on damaged by electron beam at high dose (at 70 pA). In contrast, the atomic structure of monolayer MoS₂ remains intact in low dose mode (at 15 pA). Therefore, all HAADF and DPC data of our prepared monolayer MoS₂

were collected at a dose of 15 pA, and the optimized electron beam dose did not cause any new defects.

For the revision, the following discussion is added on Page 20 of the main text: “The AC-STEM characterization was performed using a ThermoFisher Themis Z microscope equipped with two aberration correctors under 300 kV and 15 pA”.

Figure R4. (a) HAADF image of monolayer MoS₂ at low magnification; (b,c) atomic-resolution HAADF images were collected at different currents 15 pA and 70 pA, respectively.

Comment:

4. The creation of electrodes to carry out 3-electrode electrochemical measurements is certainly a commendable job. The device fabrication involves spincoating and liftoff. Doesn't the fabrication process alter the surface potentials/state of defects in the MoS₂. Wouldn't the creation of electrodes first on the substrate and transferring the MoS₂ later by a transfer method (Schneider, Grégory F., et al. "Wedging transfer of nanostructures." Nano letters 10.5 (2010): 1912-1916) give a more accurate estimation of surface potentials in the electrochemical measurements?

Response:

Thank you very much for your correct and thoughtful question. Firstly, the surface potentials/states of defects in MoS₂ can indeed be interfered with by various external factors, leading to inaccuracies in various optical, electrical, magnetic as well as electrocatalytic performance. These external influences come from the adsorption

of active gases, small molecules and small impurities in the air, as well as residual polymers from photoresist et al. However, unlike three-electrode electrochemical measurements, the all 2D MoS₂ samples for STEM and DPC tests in this paper are directly transferred to the copper microgrid, which will avoid the defect structure damage caused by 2D surface stress or electron beam in device preparation. Therefore, such antisite defects can only come from the defects constructed by our annealing process.

Meanwhile, in this paper, the construction of micro-electrocatalytic devices will inevitably produce tiny residual polymer pollutants. Moreover, these polymer residues can change the surface potential of MoS₂, thus affecting the accurate evaluation of catalytic properties. However, we are based on the same fine process and device preparation technology for testing the HER performance of all 2D MoS₂ samples, which eliminates the interference of external factors caused by device preparation. It should also be noted here that the micro-electrochemical device process we prepared is carried out based on predecessors (*Nat. Commun.* 13, 3063 (2022); *Nat. Catal.* 5, 212–221 (2022); *Nat. Mater.* 18, 1098–1104 (2019)). We explored a strict process to ensure that the 2D sample will not be damaged during the lithography process, as well as obvious surface pollution. However, our device manufacturing process or the creation of electrodes first on the substrate and transferring the MoS₂ later (*Nano letters* 10.5 (2010): 1912-1916) both require electron beam to exposure an electrochemical window to be tested in order to contact the electrolyte to form a three-electrode test system. The defect damage caused by residual glue and stress from these two technologies cannot be effectively avoided, and causes some changes in the surface potential of monolayer MoS₂.

Compared with pristine MoS₂, the improvement of HER performance of the MoS₂ with antisite defects can only come from the antisite defect active sites constructed by our annealing process. Moreover, all the 2D samples in this paper are under the same device preparation process. Through the horizontal comparison of HER performance, we can see that such HER performance changes caused by device preparation will occur in all the 2D samples, and will cancel when compared to each other. The difference in HER catalytic performance with or without defects can only come from the surface potential difference rooted in the antisite defect as well as the

performance improvement derived from the electric field polarization described in this paper.

In order to further in-situ evaluate the HER performance changes of the antisite defect structure, and confirm that the improved HER performance only comes from artificially created defect structures, we will use the in-situ liquid phase electron microscopy technology in the future work to evaluate the difference in surface structure and electric field polarization of this defect in the process of hydrogen evolution under real conditions. Thus, the intrinsic reason of the HER performance difference caused by the antisite defect structure can be directly observed and verified.

Comment:

5. How reproducible are the electrochemical measurements? Are the Tafel slopes indicating a direct correlation with the defects based on the synthesis procedure, or is there an influence of the device fabrication method of spin-coating and lift-off and subsequent processes?

Response:

We appreciate the suggestion from the reviewer. First of all, we conducted several electrocatalytic HER tests on 2D S_{2Mo}-MoS₂-5 samples. As shown in **Figure R5**. The five S_{2Mo}-MoS₂-5 samples with antisite defects basically maintained relatively stable in the HER properties, especially through the comparison of polarization curves and Tafel slopes. The results show that the current density and the Tafel slope are basically the same, indicating that the homogeneity of all samples is good, and our electrochemical performance test is repeatable.

Meanwhile, in order to compare the microscopic changes in the surface structure of 2D MoS₂ before and after the micro-electrochemical device fabrication (such as spin-coating and lift-off), Raman and PL spectra were used to characterize five different monolayer 2D MoS₂ samples with antisite defect. As shown in **Figure R6**, the Raman and PL spectra of the five samples basically remained unchanged, indicating no obvious additional pollution and no significant new structural defects during the device fabrication process.

It should also be noted here that the micro-electrochemical device process we

prepared is carried out based on predecessors (*Nat. Commun.* 13, 3063 (2022); *Nat. Catal.* 5, 212 (2022); *Nat. Mater.* 18, 1098 (2019)). In addition, the damage caused by residual glue and stress in the process of device fabrication cannot be effectively avoided, and causes some changes in the surface potential of MoS₂. However, the photoresist contamination of device fabrication has little effect on the performance of HER tested by micro-electrochemical device. The obvious changes in HER performance (Tafel slopes) are mainly caused by antisite defects in during the synthesis of monolayer S₂Mo-MoS₂.

The related results of **Figure R5** has been added as Supplementary Fig. 13 in Page 14 of the revised SI file. For the revision, the following discussion is added on Page 11 of the main text: “Furthermore, multiple S₂Mo-MoS₂-5 samples were conducted to electrocatalytic HER tests (Supplementary Fig. 13). The results show that the HER performance remained relatively stable, which illustrates the rough homogeneity of the number of antisite defects in S₂Mo-MoS₂-5 samples”.

Figure R5. (a) Polarization curves and (b) Tafel plots of five 2D S₂Mo-MoS₂-5 samples. The results show that the current density and Tafel slope are basically the same.

Figure R6. (a) Raman and (b) PL spectra of the five monolayer 2D S_{2Mo} - MoS_2 samples before and after device preparation.

Comment:

6. Overall, the authors observe a variation in the surface potential due to the presence of defects clearly from electrochemical measurements. They also simulate the same by introducing defects. If they manage to make it a little more clearer in the written text or support with an additional simulation that the surface potential may alter in presence of electrolyte and in addition introducing defects alters further and if they manage to establish a link with the electrochemical experimental observation, this would make the outcome even more stronger. Nonetheless, not having this strictly doesn't demerit any of the observations in this study.

Response:

We appreciate the suggestion from the reviewer. Indeed, we observed a variation in the surface potential and electric field polarization due to the introduction of antisite defects from both theoretical calculations and DPC electron microscopy technology, which is also the main reason for the improvement in HER properties proposed in this manuscript. We also tried additional surface potential simulations to prove this conclusion in the presence of electrolyte and in addition introducing defects. Similar to comment #1, to elucidate the impact of the antisite defects on the surface potential, this distribution of surface electrostatic potential was computed to more intuitively investigate the influence of antisite defects on the surface electric field potential under experimental conditions (**Figure R1**). In comparison to the defect-free MoS₂ at a significant separation, there is a significant variation in the potential at defect sites. This observation aligns well with the theoretical proposition that regions with lower electrostatic potential are more prone to electron donation.

Meanwhile, we added the discussion about the surface potential change in the text of the manuscript and SI, which also revealed the change in the surface potential caused by the defect more clearly. However, due to our lack of strong mathematical skills, it is difficult to establish a mathematical model corresponding to the change of surface potential or electric field polarization and HER properties. We will continue to strive to provide more favorable and direct mathematical models and theories for general scientific research to support the development of the electrocatalysis field.

Suggestions on rephrasing the sentences:

If I understand what the authors are trying to communicate, my advice is to simply a few of these sentences. Apart from these modifications, there are several typographical errors and misplacing articles which may be checked at the end.

1. Please consider rewriting the first few sentences in the abstract as: “The thriving field of atomic defect engineering towards advanced electrocatalysis relies on the critical role of electric field polarization at the atomic scale. While this is proposed theoretically, the spatial configuration, orientation, and correlation with specific

catalytic properties of materials are yet to be understood.” Instead of the present text

Response:

We appreciate the suggestion from the reviewer. We have modified the sentences in Abstract of this article. For the revision, the following sentences is added on Page 2 in the Abstract: *“The thriving field of atomic defect engineering towards advanced electrocatalysis relies on the critical role of electric field polarization at the atomic scale. While this is proposed theoretically, the spatial configuration, orientation, and correlation with specific catalytic properties of materials are yet to be understood.”*

2. Please rephrase the following sentence: “As such, analysing the non-periodic....” as: “Analyzing the non-periodic electric field on the electrocatalyst with high spatial accuracy is crucial for understanding the catalytic mechanism. Unfortunately, since the technical challenges of atomic imaging currently hinder the characterization of the non-periodic electric fields surrounding specific atomic defects, understanding such microscopic mechanisms largely relies on theoretical calculations¹⁴⁻¹⁶.”

Response:

We appreciate the suggestion from the reviewer. We also have rephrased the sentences in this article. For the revision, the following sentences is added on Page 3 of the main text: *“Analyzing the non-periodic electric field on the electrocatalyst with high spatial accuracy is crucial for understanding the catalytic mechanism. Unfortunately, since the technical challenges of atomic imaging currently hinder the characterization of the non-periodic electric fields surrounding specific atomic defects, understanding such microscopic mechanisms largely relies on theoretical calculations¹⁴⁻¹⁶.”*

3. The sentence “Therefore, MoS₂ with point defects...” may be simplified as: “Therefore, MoS₂, with point defects as a catalyst, is an ideal material system to explore the effect of electric field polarization of atomic defect sites on the as-tuned catalytic property and performance.”

Response:

We appreciate the suggestion from the reviewer. We have modified the sentences in this article. For the revision, the following sentences is added on Page 4 of the main text: *“Therefore, MoS₂, with point defects as a catalyst, is an ideal material system to explore the effect of electric field polarization of atomic defect sites on the as-tuned catalytic property and performance.”*

4. The sentence “In fact, the asymmetric charge density distribution....” May be rephrased as: “The asymmetric charge density distribution on the catalyst surface is directly related to its electric field, which is the decisive origin of catalytic performances^{6,11-16,26}. Here, the DPC technology, a recent advancement in STEM imaging²⁷⁻³⁰, was selected to characterize the electric field distributions of the MoS₂-based materials.”

Response:

We appreciate the suggestion from the reviewer. We have modified the sentences in this article. For the revision, the following sentences is added on Page 15 of the main text: *“The asymmetric charge density distribution on the catalyst surface is directly related to its electric field, which is the decisive origin of catalytic performances^{6,11-16,26}. Here, the DPC technology, a recent advancement in STEM imaging²⁷⁻³⁰, was selected to characterize the electric field distributions of the MoS₂-based materials.”*

5. With the experimental results presented in this study, the authors may go as far as making a stronger conclusion by modifying the last sentence in conclusions as: “This is the first example to explore the relationship between the atomic-level polarization in electric fields of catalyst surfaces and their catalytic activity. This study not only correlates the influence of atomic-level defects on catalytic activity but also paves the way to potential real-time catalysis studies using MEMS-based devices in microscopes.”

Response:

We appreciate the suggestion from the reviewer. We have modified the sentences in Conclusions of this article. For the revision, the following sentences is added on Page 19 of the main text: *“This is the first example to explore the relationship between the atomic-level polarization in electric fields of catalyst surfaces and their catalytic activity. This study not only correlates the influence of atomic-level defects on catalytic activity but also paves the way to potential real-time catalysis studies using micro-electro mechanical systems (MEMS)-based devices in microscopes.”*

Responses to the comments of Reviewer #4:

Comment:

Reviewer #4: In this study, Xu et al. have undertaken a comprehensive investigation of electric field polarization and its impact on the hydrogen evolution reaction (HER) efficiency in defective MoS₂. Their work combines experimental techniques and theoretical calculations to shed light on the factors influencing catalytic activity. While the experimental section is commendable, there are some suggestions for refining the theoretical portion of the study before it can be considered for acceptance.

1. Figure 3d should include labels for the hydrogen adsorption free energies, even though this information is available in the supplementary material. This will enhance the clarity of the figure and make it easier for readers to interpret.

Response:

Thank you very much for reviewing our manuscript. We greatly appreciate your inspiring, useful, helpful, and constructive comments. Indeed, without labels for the hydrogen adsorption free energies, we would only be able to see a rough trend and not be able to precisely determine the specific catalytic properties of each structure. As suggested by the Reviewer, we have added labels for the hydrogen adsorption free energies according to the detailed ΔG_{H^*} values in the supplementary material and used different colors to distinguish pristine MoS₂, V_{Mo}-MoS₂, and S_{2Mo}-MoS₂, as shown in **Figure R1**.

On page 12 of the revised manuscript file, **Figure R1** has been added to Figure 3 as Figure 3d to replace the previous HER free-energy diagram for pristine MoS₂, V_{Mo}-MoS₂, and S_{2Mo}-MoS₂. In addition, the Figure note for Figure 3d has been revised to read as follows: "*HER free-energy diagram with detailed ΔG_{H^*} values for pristine MoS₂, V_{Mo}-MoS₂, and S_{2Mo}-MoS₂*".

For your convenience, all changes have been highlighted by yellow in the revised manuscript and Supplementary Information (SI) file.

Figure R1. HER free-energy diagram with detailed ΔG_{H^*} values for pristine MoS₂, V_{Mo}-MoS₂, and S₂Mo-MoS₂.

Comment:

2. It is better to provide a quantitative explanation for the observed hydrogen adsorption energies in the context of electronic structure, possibly using theories such as band center theory. This would help readers better understand the underlying mechanisms driving the variations in adsorption energies for V_{Mo}-MoS₂, S₂Mo-MoS₂, and MoS₂.

Response:

Thank you for this constructive suggestion. According to the Reviewer's suggestions, we also tried to use band center theory to explain the variations in observed hydrogen adsorption energies for V_{Mo}-MoS₂, S₂Mo-MoS₂, and MoS₂. Unlike the d-band center theory for metal atoms, since the active site of HER is a non-metallic S atom with no d-band, we mainly calculate the p-band center of the active S atom here. According to the previous works (*J. Mater. Chem. A*, 2020, 8, 5688; *Carbon* 2018, 133, 260), the p-band center of active S sites can be calculated using:

$$\varepsilon_p = \frac{\int_{-\infty}^0 ED(E) dE}{\int_{-\infty}^0 D(E) dE}$$

where $D(E)$ is the density of states projected onto the p-orbital of active S atoms and E is the energy position of p-orbital relative to the Fermi level.

We mainly calculated the p-band centers of the active S atoms on pristine MoS₂ and V_{Mo}-MoS₂ as well as the three better S active sites in the S_{2Mo}-MoS₂, as shown in **Figure R2**. The calculated p-band centers aligned at the Fermi energy level are displayed in **Figure R3**. In addition, it is also important to note that pristine MoS₂, V_{Mo}-MoS₂, and S_{2Mo}-MoS₂ exhibit distinct work functions (the difference between the vacuum energy level and the Fermi energy level) due to the large structural differences, as shown in **Figure R4a**. Therefore, in **Figure R4b**, we further corrected the p-band center with the work function alignment method to ensure that all the values of the p-band center are relative to the uniform vacuum energy level. **Figure R4c** shows the p-band center as a function of the hydrogen adsorption energies ΔG_{H^*} on pristine MoS₂, V_{Mo}-MoS₂, and S_{2Mo}-MoS₂. Unfortunately, there is not a clear relationship between the hydrogen adsorption energies and the p-band center, suggesting that the band center theory can not provide a quantitative explanation for the variations in hydrogen adsorption energies for our investigated systems. We have also referred to several articles about HER calculations on the MoS₂-based systems (*Nat. Mater.* 2016, 15, 48; *J. Am. Chem. Soc.* 2020, 142, 4298; *ACS Catal.* 2021, 11, 4486) but found nothing that explains the trend of hydrogen adsorption energy using band center theory.

In addition, we also try to explain the variations of the observed hydrogen adsorption energies from the coordination bonding point of view. We all know that for perfect pristine MoS₂, the surface S atoms are bonded to Mo atoms in three Mo-S bonds, making the S atoms present a nearly saturated bonding state, as shown in **Figure R5a**. Therefore, the adsorption of additional H* on the S atom is then more difficult, accompanied by a very positive hydrogen adsorption energy of $\Delta G_{H^*} = 2.12$ eV. To improve the HER catalytic activity, it is necessary to break the original stable S coordination environment to enhance the hydrogen adsorption energy. For V_{Mo}-MoS₂ in **Figure R5b**, the absence of the Mo atom allows the S atom around the V_{Mo} to retain only two Mo-S bonds, so the unsaturated S atom will then have a relatively strong adsorption with H* of $\Delta G_{H^*} = -0.41$ eV. In **Figure R5c-d**, it can be seen that two-coordinated S atoms with two Mo-S bonds also appear in S_{2Mo}-MoS₂ structures, but unlike in V_{Mo}-MoS₂, the two-coordinated S atoms in S_{2Mo}-MoS₂ structures have more pronounced structural distortions. Highly symmetric structures tend to have lower

energies than those with structural distortions. Thus, despite both being two-coordinated S atoms, structural distortions of S_{2M_0} - MoS_2 will weaken the adsorption of H^* compared to symmetric two-coordinated S atoms in V_{M_0} - MoS_2 . In addition, in S_{2M_0} - MoS_2 structures, the antisite defects lead to the formation of a special three-coordinated S atom possessing two Mo-S bonds as well as one S-S bond, as in the S site labelled 2 in **Figure R5d**. The S-S bond is relatively weak compared to the Mo-S bond, so compared to the three-coordinated S atoms in pristine MoS_2 with three strong Mo-S bonds, the weak S-S bond in S_{2M_0} - MoS_2 will allow the special 3-coordinated S atom to have a greater redundancy of bonding energy to adsorb H^* , thus enhancing the hydrogen adsorption. However, the special 3-coordinated S atom in S_{2M_0} - MoS_2 has one more S-S bond than the 2-coordinated S atom in V_{M_0} - MoS_2 , thus weakening the H^* adsorption. In summary, coordination bonding-based analysis can elucidate the variations in hydrogen adsorption energies for pristine MoS_2 , S_{2M_0} - MoS_2 , and V_{M_0} - MoS_2 .

On page 16 of the revised SI, **Figure R5** has been added as Supplementary Fig. 15. Additionally, the related discussion has been added on Pages 13 and 14 of the revised manuscript file and is highlighted in yellow color, as follows: *“Furthermore, the coordination bonding-based analysis is used to elucidate the variations in hydrogen adsorption energies. For pristine MoS_2 , the nearly saturated bonding state of S atoms makes the adsorption of additional H^* difficult (Supplementary Fig. 15a), resulting in weak hydrogen adsorption. For V_{M_0} - MoS_2 in Supplementary Fig. 15b, the introduction of V_{M_0} leads to two-coordinated S atoms with two Mo-S bonds, resulting in an unsaturated property that causes relatively stronger H^* adsorption. In Supplementary Fig. 15c-d, unlike in V_{M_0} - MoS_2 , the two-coordinated S atoms in S_{2M_0} - MoS_2 structures exhibit more pronounced structural distortions, which will weaken the adsorption of H^* compared to the symmetric two-coordinated S atoms in V_{M_0} - MoS_2 . Additionally, for the unique three-coordinated S atom with two Mo-S bonds and one S-S bond, the weaker S-S bond than the Mo-S bond allows for a greater redundancy of bonding energy to adsorb H^* than pristine MoS_2 , thereby enhancing hydrogen adsorption. In contrast, the additional S-S bond would result in much weaker adsorption compared to two-coordinated S atoms in V_{M_0} - MoS_2 . Overall, the formation of antisite defects breaks the original coordination situation of active S atoms and introduces new coordination environments more favorable for H^* adsorption than pristine MoS_2 and*

$V_{\text{Mo}}\text{-MoS}_2$, thus improving the catalytic activity”.

Figure R2 The atomic structures of pristine MoS_2 , $V_{\text{Mo}}\text{-MoS}_2$, and antisite defect structures of $\text{S}_{2\text{Mo}}\text{-MoS}_2@5\text{S}$ and $\text{S}_{2\text{Mo}}\text{-MoS}_2@4\text{S}$, in which @5S and @4S denote the number of S atoms around antisite defect. The detailed ΔG_{H^*} values are also shown and active sites are denoted by red circles, which correspond to Fig. 3d in the manuscript file.

Figure R3 Projected density states of the p-orbital of the active S atoms on pristine MoS_2 , $V_{\text{Mo}}\text{-MoS}_2$, and antisite defect structures of $\text{S}_{2\text{Mo}}\text{-MoS}_2@5\text{S}$ and $\text{S}_{2\text{Mo}}\text{-MoS}_2@4\text{S}$. The p-band centers (ϵ_p) aligned at the Fermi energy level and the Fermi energy levels are denoted by orange lines and gray dashed lines, respectively. The insets display the active S atoms, indicated by red circles.

Figure R4 (a) The work function of pristine MoS₂, V_{Mo}-MoS₂, and S_{2Mo}-MoS₂. (b) The p-band center of the active S atoms using the work function alignment method. (c) The p-band center as a function of the hydrogen adsorption energies ΔG_{H^*} at the active S sites in Figure R2.

Figure R5 (a-d) The coordination demonstration of active S sites of pristine MoS₂, V_{Mo}-MoS₂, and S_{2Mo}-MoS₂.

Comment:

3. To improve the clarity of the charge density diagrams, it is recommended to include a color bar that demonstrates charge accumulation and depletion. This will make it easier for readers to interpret the diagrams and understand the spatial distribution of charges.

Response:

Thank you for your helpful comment. Following the Reviewer's suggestions, we have added the color bar in the figures (Fig. 14 in the previous Supplementary Information and Fig. 3f-I in the previous manuscript) about charge density distributions to demonstrate the charge accumulation and depletion. The revised charge density diagrams with the color bar can be found in **Figure R6** and **Figure R7**.

On page 12 of the revised manuscript file, **Figure R6** has been added as Fig. 3f-i. Also, on page 18 of the revised SI, **Figure R7** has been added as Supplementary Fig. 17. In addition, we also added a sentence in the figure note for Fig. 3f-i and Supplementary Fig. 17 to clarify the meaning of the different colors representing charge accumulation and depletion, as follows: "By definition, a region with a value of 1.0 denotes ideal

charge accumulation, whereas a region with a value near 0.0 signifies a remarkably low charge density.”

Figure R6 (a-d) Charge density distributions for pristine MoS₂, V_{Mo}-MoS₂, S_{2Mo}-MoS₂, and S_{Mo}-MoS₂. By definition, a region with a value of 1.0 denotes ideal charge accumulation, whereas a region with a value near 0.0 signifies a remarkably low charge density.

Figure R7 Density of states (DOS) and corresponding charge density distributions for pristine MoS₂, V_{Mo}-MoS₂, S_{2Mo}-MoS₂, S_{Mo}-MoS₂, and antisite defect structures with S vacancies. By definition, a region with a value of 1.0 denotes ideal charge accumulation, whereas a region with a value near 0.0 signifies a remarkably low charge density.

4. In the Methods section, it is imperative to explicitly mention the use of the computational hydrogen electrode (CHE) method for calculating Gibbs free energy instead of simply using 1/2 H₂ directly. Additionally, cite the relevant literature, such as J. Phys. Chem. B 2004, 108(46), 17886–17892 and J. Electrochem. Soc. 2005, 152 (3), J23-J26. by J. K. Nørskov, to provide a reference for this approach, enhancing the transparency of the computational methodology.

Response:

Thank you for your helpful and constructive comment. We do apologize for ignoring these citations to the relevant literature. According to the Reviewer's suggestions, we have carefully read these highly important papers and cited them in the revised manuscript file. On page 31 of the revised manuscript file, [*J. Phys. Chem. B* 2004, 108(46), 17886–17892; *J. Electrochem. Soc.* 2005, 152 (3), J23-J26] have been cited as Ref. 42 and Ref. 43, respectively.

5. Listing the detailed calculation parameters in the climbing image nudged elastic band (cNEB) method would be better.

Response:

Thank you for your helpful comment. As suggested by the Reviewer, in the DFT calculations section on page 23 of the revised manuscript file, we have added the corresponding discussion to show the detailed calculation parameters in the cNEB method, which are highlighted in yellow color, as follows: *".....As in the structural optimization above, the same cut-off energies of 450 eV, the k-point mesh of 4 × 4 × 1, and convergence criteria are used for the supercell with 5 × 5 × 1 surface periodicity*

during the cNEB calculations. Between the stable initial and final states, we inserted three points to determine the optimal growth paths.”

6. Did the authors include the Van der Walls interactions and solvation correction in the DFT calculations?

Response:

Thank you for this constructive suggestion. In our DFT calculations, we did not employ the Van der Walls interactions and solvation correction. In our preparation of the manuscript, we mainly referred to several HER-related papers (*Nat. Energy*, 2016, 1, 1; *J. Am. Chem. Soc.* 2020, 142, 4298), which also do not include the Van der Walls interactions and solvation correction. Usually, we consider the Van der Walls interactions in the presence of atoms that are not bonded between the adsorbate and the substrate. For instance, in **Figure R8a**, during the oxygen reduction reaction (ORR) process, the OOH* intermediate state only has one O bonded to the substrate, while the remaining OH suspended on the surface of the substrate is not bonded to the substrate. Hence, Van der Waals interactions are required in this scenario. However, for the HER reaction in **Figure R8b**, only H* is adsorbed on the substrate, and H* is fully bonded to the substrate, so Van der Waals interactions are not needed at this point. Therefore, we do not employ Van der Walls interactions and solvation correction in our calculations.

In addition, we also examined the effect of solvation correction on the hydrogen adsorption energy of pristine MoS₂, V_{Mo}-MoS₂, and S_{2Mo}-MoS₂ with the optimal HER activity. As shown in **Figure R9**, we mainly adopted explicit models to tackle solvent effects, in which multiple water molecules are added to the catalyst surfaces to model the aqueous interface. The DFT-D3 scheme is used to describe the van der Waals interactions between the adsorbed H* and the water molecule layer. **Figure R9d,e** show the HER free-energy diagram without and with solvent correction, respectively. The solvation correction has little influence on the hydrogen adsorption energy, suggesting that the solvation correction is negligible and does not affect the fact that the S_{2Mo}-MoS₂ has a superior HER activity over pristine MoS₂ and V_{Mo}-MoS₂.

Figure R8 (a) Atomic structures of reaction intermediates OOH* in oxygen evolution/reduction reactions. (b) Atomic structures of adsorbed H* on pristine MoS₂.

Figure R9 (a-c) Atomic structures of substrates and adsorbed H* with water molecules layer on pristine MoS₂, S₂Mo-MoS₂, and V_{Mo}-MoS₂. (d-e) HER free-energy diagrams for pristine MoS₂, S₂Mo-MoS₂, and V_{Mo}-MoS₂ without and with solvation correction.

Overall, these revisions will strengthen the theoretical aspect of the paper and improve its accessibility to the scientific community.

REVIEWERS' COMMENTS

Reviewer #1 (Remarks to the Author):

I have carefully read the revised manuscript and response letter, and satisfied their changes. I think the manuscript can be published in current version.

Reviewer #2 (Remarks to the Author):

The suggested changes have largely been implemented and in my view the Manuscript reads well now in my opinion.

I would like to compliment the team of authors once again for bringing out this very nice study.

Just one suggestion, in the caption for Supplementary Fig. 13, please change: "The results show that the current density and Tafel slope are basically the same", to " The results show that our electrochemical measurements are reproducible from the similar current density values and slopes."

Reviewer #4 (Remarks to the Author):

The authors have adequately addressed all concerns raised in the initial review. The manuscript is now suitable for publication.

Responses to the Comments

Responses to the comments of Reviewer #1:

Comment:

Reviewer #1: I have carefully read the revised manuscript and response letter, and satisfied their changes. I think the manuscript can be published in current version.

Response:

We appreciate the reviewer for the very positive recommendation of our work. Thank you very much for your professional and kind suggestions, which have helped us improve the integrity of the manuscript.

Responses to the comments of Reviewer #2:

Comment:

Reviewer #2: The suggested changes have largely been implemented and in my view the Manuscript reads well now in my opinion.

I would like to compliment the team of authors once again for bringing out this very nice study.

Just one suggestion, in the caption for Supplementary Fig. 13, please change: "The results show that the current density and Tafel slope are basically the same", to "The results show that our electrochemical measurements are reproducible from the similar current density values and slopes."

Response:

We appreciate the reviewer for the very positive recommendation of our work. We also have modified the sentences in this article. For the revision, the following sentences is added on Page 14 of the Supplementary Information file: *"The results show that our electrochemical measurements are reproducible from the similar current density values and slopes."* Thank you again for your professional suggestions.

Responses to the comments of Reviewer #4:

Comment:

Reviewer #4: The authors have adequately addressed all concerns raised in the initial review. The manuscript is now suitable for publication.

Response:

We appreciate the reviewer for the very positive recommendation of our work. Thank you again for your professional suggestions.